# Heavy Metal Analysis and Health Risk Assessment of Potato (*Solanum tuberosum* L.) Cultivars irrigated with Fly Ash-Treated Acid Mine Drainage

**Maropeng Vellry Raletsena [1,*]**, **Nkoana Ishmael Mongalo [1]** and **Rabelani Munyai [2]**

[1] College of Agriculture and Environmental Sciences (CAES), University of South Africa, Private Bag X6, Florida 1710, South Africa

[2] Department of Agriculture and Animal Health, College of Agriculture and Environmental Sciences (CAES), Horticulture Centre, University of South Africa, Private Bag X6, Florida 1710, South Africa

* Correspondence: raletmv@unisa.ac.za; Tel.: +27-116709433

**Abstract:** In water-scant regions, the reuse of (un)treated acid mine drainage effluent (AMD) water for crop irrigation has turned into a prerequisite. The study assesses the levels of heavy metals, and health risk assessment in two potato crop cultivars, namely, *Fianna* and *Lady rosetta* (both determinate and indeterminate) when exposed to irrigation with different fly ash: acid mine drainage amelioration ratios. The study investigates the health risk assessment in the potato tissues namely, stem, tubers, new and old leaves of the potato cultivars. The treatments constituted a control, 50% FA: AMD, 75% FA: AMD ratio, and 100% AMD (untreated AMD). The results showed that the heavy metals of plants irrigated with AMD mixed with FA was significantly affected differently at harvest. In summary, the concentration of Cd was over the WHO acceptable standards in untreated acid mine drainage water. *Fianna* recorded 1.34 mg/kg while a notably decrease was observed in *Lady rosetta* with (1.01 mg/kg). In any case, FA was proven to significantly lessen the Cd particles in both FA-AMD ratios: Cd content was 0.98 mg/kg and 0.84 mg/kg in 50% FA: AMD treatment for *Fianna* and *Lady rosetta* separately, while the recommended maximum limit is 0.1 mg/kg. With the readings recorded, they are slightly high according to the CODEX general standard for food contaminants and toxins in food and feed (CODEX STAN 193-1995). Then again, in 75% FA: AMD ratio, the Cd concentration was 0.04 mg/kg and 0.03 mg/kg for *Fianna* and *Lady rosetta* exclusively. It is in this way obvious that FA adsorbed the Cd ions, and the two cultivars fulfill the CODEX guideline, nonetheless the potato crop irrigated with a 75% FA: AMD ratio can be consumed by humans without causing any detrimental effects.

**Keywords:** potato; fly ash (FA); acid mine drainage (AMD); estimated daily intake (EDI); target cancer risk (TR); target hazard quotient (THQ); bioaccumulation factor (BAF); translocation factor (TF)

## 1. Introduction

Freshwater supply is one of the world's greatest issues, with roughly 33% of the world's drinking water coming from surface sources like streams, dams, lakes, and trenches. South Africa, in the same way as other different nations, contamination of existing water assets is the most risk to a reasonable water supply [1]. The country is recorded as one of the driest countries on the planet, and with a serious water shortage projected very soon [2]. High populaces development rate, industrialization, and urbanization has brought about a worldwide deficiency of great water supplies [3–6]. Rural and urban populaces depend on surface water, and with the country's economy vigorously dependent on mining, the water supplies are logically being contaminated, strikingly by corrosive acid mine drainage effluent (AMD) [5].

Corrosive mine seepage is an overall result of mining tasks [6]. Mine profluent, which is both synthetically harmful and radioactive (on account of the West Rand), is

delivered in different ways by mines, including run-off from mine dumps into surface and groundwater, leakage from mine dumps into underground water, and flood from deserted mines [7]. The way that AMD is incredibly hard to ameliorate and can persevere for a really long time, even after mine conclusion, as demonstrated by [8], makes it remarkable and particularly problematic. Changes in the quality and amount of irrigation water system water have different hindering outcomes on irrigated agricultural output, including diminished yields, deteriorated quality, and soil quality corruption [9]. Scientists [6–10] have recommended utilizing lime, phosphorous stone, fake wetlands, and the converse assimilation methodology to further develop corrosive mine seepage so it tends to be utilized as irrigation water for crop production.

The cost of decontaminating AMD water varies, with some treatments, such as fly ash (FA), being less expensive and capable of increasing irrigation water supply compared to liming, which is costly [10]. Because it is less expensive, using FA to cure AMD could help reduce water constraints in the South African agriculture business. In fact, an earlier study has revealed that FA could be a viable alternative to liming in the treatment of AMD [11]. While the rest of the world has made significant progress in adopting environmentally friendly technology to reduce mining pollution and alleviate poverty, Africa, particularly Sub-Saharan Africa, has lagged [6–10]. Even though the country's economy is fueled by strong mining industry, South Africa is beset by a water security dilemma [11]. These two phenomena have harmed populations and ecosystems as a result of highly acidic water pouring into the country's water supply [9–11]. The contaminated water released by abandoned mines is a source of concern for surrounding residents, especially those who live along the Vaal and Limpopo rivers [12]. This alarming situation, combined with the threat of climate change to food security, mandates the deployment of less costly, widely available, and environmentally friendly alternatives to alleviate the country's dilemma. This study aims to determine if AMD is suitable for irrigation of potatoes, as well as the effects that this type of water has on the crop's biochemical composition and physiological features, as well as its rhizosphere.

Although there are various acid mine water treatment techniques and procedures available, they all have drawbacks, such as high costs and difficulties with large precipitation resulting in insolubility [9,10]. Phosphate rock, for example, has been used to treat AMD in several studies. Phosphate is usually much more expensive than other calcium-based supplements, yet it is required in similar proportions [11]. However, due to its low solubility and inclination to form an exterior coating or armor of ferric hydroxide (Fe(OH)3) when added to AMD, the successful application of limestone is limited [12,13]. Wetlands, on the other hand, are frequently suggested as a way to reduce heavy metals in contaminated water. A wetland system has three basic processes: (1) soil and substrate, (2) hydrology, and (3) plants. Because the entire process is interdependent, the heavy metal removal mechanisms in wetlands are quite intricate, and heavy metals have a deleterious impact on plants [14].

According to Yunusa et al. [15], fly ash can be used as an adsorbent to remove common dyes from acid mine drainage. Many potential positive applications of fly ash have been investigated in order to reduce waste, lower disposal costs, and generate value-added products [16]. It was shown to be useful as an ameliorant because it could restore the physical, chemical, and biological aspects of problematic soils while still containing easily available macro-and micronutrients for plant uptake. The use of fly ash in damaged soil increases plant biomass output, according to studies [15,16]. FA is a good soil ameliorant, according to the researchers, and its qualities boost soil fertility and productivity. Rios et al. [17,18] are among the researchers that have conducted extensive investigations that show fly ash may be used successfully to purify water and remove heavy metals from polluted water. They also confirmed that it is an important source of plant nutrients (e.g., Ca, Mg, K, P, S, B, Fe, Cu, and Zn).

Fly ash materials have also been considered as a viable resource for agricultural and industrial purposes. In agriculture, for example, it is valued as a useful and non-toxic

fertilizer or soil amendment that can be used to clean up the soil [15]. This idea has been supported by a number of other scholars [16]. The characteristics are important since there are concerns about future lime supplies, particularly that deposits may be depleted by 2050 [16]. When more salt is added to water, the pH changes from neutral to alkaline [17,18]. This principle is applicable in the current study, both in terms of AMD treatment and the usage of fly ash. Heavy metals can form solutions in water, according to Nemutanzhela et al. [19], even if some of them are dangerous while in solution. Water dilution, according to [20], is founded on the idea that when water is added to specific elements, like as sodium (Na), NaOH is created based on the concentration of Na ions.

The research provided above refers to a variety of ways that have been employed in the past to mitigate the negative effects of AMD water; however, none of the solutions entail treating AMD in order to evaluate its usage in crop production. This study offers combining AMD water with varying quantities of FA. This approach has received little, if any, research and is not currently described in the literature. The harmful effects of AMD are expected to be decreased by using the water dilution strategy, and the product of this approach can be utilized to irrigate crops that are suitable for human consumption and have minimum environmental impact. Acid mine drainage contains substantial amounts of heavy metals, according to a recent study [21], and fly ash was employed to reduce the concentrations. Fly ash create a method that works for increasing ammonium and phosphate removal effectiveness at low concentrations without reducing removal capacity [22]. Other studies [23,24] previously discovered higher quantities of cadmium, cobalt, copper, lithium, mercury, strontium, and zinc in rice leaves, which they felt led to the production of necrotic areas and a significant degradation of Rubisco, particularly Rubisco LSU. Researchers have long investigated nutritional and metabolomic profile (molecular) components in diverse plants [25]. There are currently no published studies on the effects of heavy metals on metabolites in potatoes (treated or untreated with fly ash).

Potatoes (*Solanum tuberosum* L.) are grown for food and are the world's fourth most significant food crop, after rice, maize, and wheat [26]. The cultivation of this crop generates a lot of economic activity all over the world, but it takes a lot of water for irrigation [27]. Despite the fact that South Africa is a water-scarce country, there are considerable areas of land dedicated to potato farming. Furthermore, in some places where additional irrigation is used for potato production, surface water is polluted by mining by-products such as acid mine drainage and/or coal-burning fly ash. Acid mine drainage, in particular, is a serious environmental issue because it contains relatively high concentrations of heavy metals (Hg, Fe, Mn, Pb, Sr, Zn, and SO4), which can persist and bio-accumulate in plant cells, causing health risks to humans and animals [28]. Fly ash, on the other hand, has been shown to reduce heavy metal concentrations in soils such as Cd, Hg, Mn, Ni, Al, Fe, and Ti [27–29]. It has been proven, for example, to reduce heavy metal toxicity in contaminated waters [29].

## 2. Materials and Methods

### 2.1. Preparation of Samples

The AMD used in the experiments was obtained from one of South Africa's gold-producing mines in Randfontein, Gauteng Province, on the other hand, the FA came from one of ESKOM's electricity-generating plants in the Mpumalanga Province. The AMD was mixed with FA (FA-treated AMD) at different ratios of 50% FA: AMD ratio, 3:1 ($v/w$) while the untreated AMD and the control (tap water) were expressed by ratios 100% FA: AMD ratio and a control respectively. Solution 50% FA: AMD was made up of 50% AMD and 50% FA, 75% FA: AMD contained 75% AMD and 25% FA. Untreated AMD, that is treatment 4, contained 100% AMD and a control contained 0% FA and tap water. Each FA: AMD irrigation (FA-treated AMD) solution was mixed in a 220L barrel. A stirrer was used to agitate the mixture for 30 min, and AMD contact time with FA was timed using a stopwatch. A bench pH meter (Model: ADWA—AD 1020, Hungary) was calibrated, and samples were measured for pH change during 6 spaced periods using [30] procedures and methodologies. The pH was assessed as impacted by FA at different time intervals

and temperatures using one sample from each container and one tap water sample taken according to well-established sampling protocols.

### 2.2. Collection and Processing of Soil and Plant Material

The study was conducted over two growing seasons (April to July 2018 (1st season) and October to December 2018 (2nd season). A growth media was prepared using the topsoil, river sand, and vermiculite at a ratio of 3:1:1 and the media was sterilized. One bag of 20 kg of 3:2:1 (24) + Zn mixed fertilizer was mixed with the growth medium. Certified seeds of two cultivars namely *Lady rosetta* (determinate) and *Fianna* (indeterminate) were obtained from First Potato Dynamics (Pty) Ltd. (Cape Town, South Africa) in the Western Cape Province, and they were stored at the National Potato Co-op (Nationale Aartappelkantoor (NAK)). The storage facilities were in Bethal in the Mpumalanga Province.

All the seed tubers were first stored at 3 °C and later transferred to 15 °C at 14-day intervals. The pot (20 × 20 cm) experiments were carried out in a greenhouse setup with each comprising of the four irrigation treatments replicated six times for each cultivar (see Figure 1). A 2 × 5 factorial experiment was laid out in a completely randomized block design in a glasshouse at the Florida campus, University of South Africa (Unisa), Johannesburg, Gauteng Province (26°10′30″ S, 27°55′22.8″ E). The experiments were carried out in two growing seasons, i.e., in autumn (Season 1) and in spring (Season 2). The treatments constituted of control, 50% FA: AMD, 75% FA: AMD ratio, and 100% AMD (untreated AMD). Plants were well irrigated before the imposition of the treatments. Irrigation treatments were executed two weeks after seedling establishment. Harvesting was done upon maturity of the plants at 90 days after planting [31].

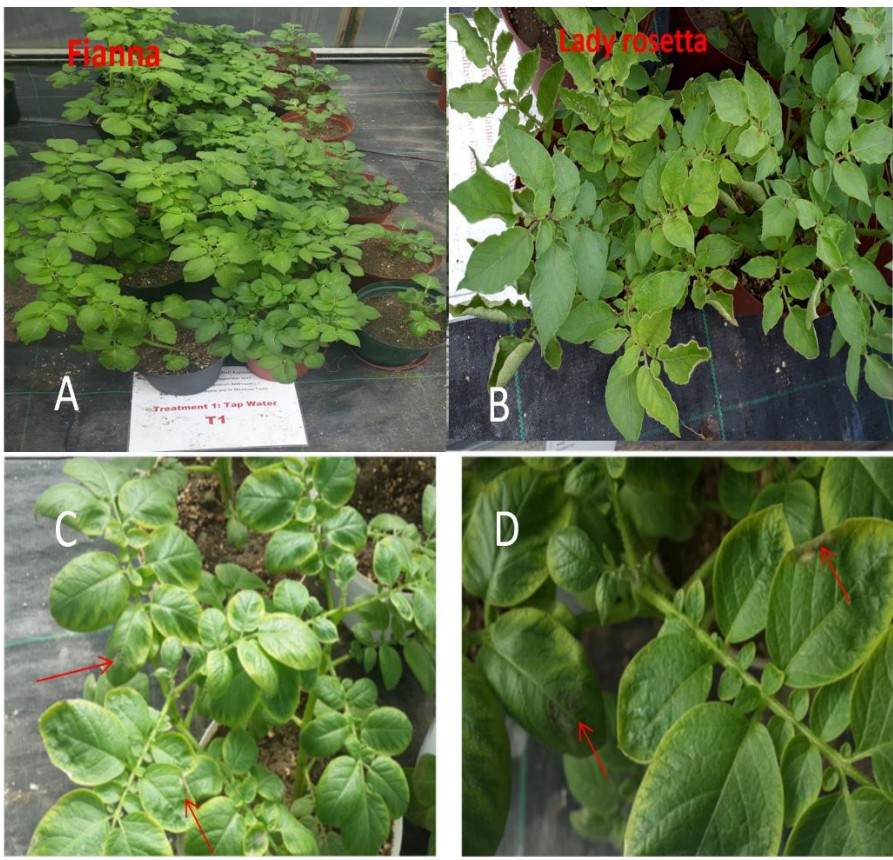

**Figure 1.** Experimental layout in greenhouse (**C**) and symptoms of metal toxicity (**D**) (initiation of discolouration in leaf margins) in both cultivars (*Fianna* (**A**) and *Lady rosetta* (**B**)), exposed to irrigation with FA-treated AMD treatments.

*2.3. Determination of Concentration of Mineral Nutrients in Tubers*

Plant samples were digested using the reference method by the EPA 3050B. About 1 g of dried potato tuber samples was weighed and milled into a medium wall digestion tube (4IS 0 mm id × 276 mmm length). About 15 mL of concentrated Nitric Acid (65%) (Merck, South Africa) was added and after 2 h, contents were placed on a cold heating (digestion) block and heated to 100 °C until fumes reduced. Thereafter, contents were re-heated in increments of 10 °C up to a maximum of 140 °C for an hour to ensure complete digestion of plant material and as shown by development of a yellow color 41. After adding about 5 drops of AR-quality hydrogen peroxide (Chem lab supplies, JHB, South Africa), the mixture was heated for 5 to 10 min, cooled, and then washed with de-ionized water into a 100 °C volumetric flask. Preparation and analysis of plant samples followed [32]. The concentrations of heavy metals in plant material were recorded on a fresh-and dry weight basis (Table 1). The supernatant was filtered into a storage vessel as described by [33]. The heavy metals were analyzed from potato tubers using an Inductively coupled Plasma Mass Spectrophotometer (ICAP 6300, Thermo Electron, London, UK) from the University of South Africa, Florida, South Africa. These ions are then separated and detected by the mass spectrometer following the method EPA 3050B procedures.

*2.4. Heavy Metal Contamination and Health Risk Assessment*

2.4.1. Bioaccumulation Factor (BAF) and Translocation Factor (TF) of Toxic Metals

The bioaccumulation factor (Table 2) is the proportion of heavy metal concentration in edible part of a plant to heavy metal concentration in a soil sample [34,35]. Correspondingly, the transfer of heavy metals from soil to plant was calculated as shown by [36]: Equation (1).

$$BAF = \frac{T_c}{S_c} \tag{1}$$

where $T_c$ is metal concentration in plant tissue (mg/g dry weight (d.w.)) and $S_c$ is the metal concentration in sediment (mg/g d.w.) [37].

Abdu et al. (2011) and Safarzadeh et al. (2013) [38,39] reported that metals translocate from soil and to plant parts, root and shoot (Table 3). The following Equation (2) was used to calculate the translocation factor thereof:

$$TF = \frac{BAF \ of \ stem \ or \ leaf}{BAF \ of \ root} \tag{2}$$

2.4.2. Estimation of Potential Health Risks from Potato Consumption

In order to estimate the potential health risks associated with the consumption of potato exposed to toxic metals, the tolerable daily intake (TDI) of toxic metals and estimated daily intake (EDI) were determined first.

2.4.3. Estimated Daily Intake (EDI)

The daily intake (Table 3) of the metals considered in this study were determined based on their mean concentration in each potato sample and the estimated daily consumption of the vegetables was reported in grams. The EDI value of each metal of interest was determined by the formula utilized by [40] with slight modifications as shown in Equation (3).

$$EDI = \frac{Ef \times ED \times FIR \times CM \times CF}{BW \times TA} \times 0.001 \tag{3}$$

where *Ef* is exposure frequency (365 day/year); *ED* is the exposure duration (65 years), equivalent to average life time [41]; *FIR* is the average food (vegetable) consumption (240 g/person/day), which were obtained from the World Health Report [42] for low fruit and vegetable intake; *CM* is metal concentration (mg/kg dry weight); *Cf* is concentration conversion factor for fresh vegetable weight to dry weight (which is 0.085) [42]; *BW* is

reference body weight for an adult, which is 70 kg [43]; *TA* is the average exposure time (65 years × 365 days) and 0.001 is unit conversion factor.

2.4.4. Risk from the Intake of Heavy Metals through Ingestion (Target Hazard Quotient, THQ)

In order to assess the non-carcinogenic human health risk from the consumption of vegetables contaminated by heavy metals, the target hazard quotient (THQ) values were estimated. The THQ values (Table 4) of the local population due to the consumption of contaminated vegetables were calculated using Equation (4) as described by [44–46].

$$THQ = \frac{EDI}{R_f D} \tag{4}$$

where *EDI* is the estimated daily metal intake of the population in mg/day/kg body weight and $R_f D$ is the oral reference dose (mg/kg/day) values for each metal of interest and as listed in Table 4. Where the value of THQ is <1, it is generally presumed to be safe for the risk of noncarcinogenic effects and if it is >1, it is supposed that there is a chance of noncarcinogenic effects with an increasing probability as the value upsurges [47–49].

2.4.5. Target Cancer Risk (TR)

It has been documented that individual health risks of the analyzed heavy metals in the same vegetable are accumulative and that is expressed as hazard index (HI). The mathematical expression of HI suggested by USEPA and referred in literature [50–52] was calculated using Equation (5).

$$HI = THQ(Cd) + THQ(Pb) + THQ(Zn) + THQ(Ni) + THQ(Cu) \tag{5}$$

where *HI* = hazard index, *THQ(Cd)* = target hazard quotient for Pb intake, and a similar notation was also used for other trace metals (Table 4).

The potential carcinogenic effect (Table 5) of metals (Cd, Pb and Ni) was further characterized by calculating the maximum allowable daily potato consumption limit (TRlim). TRlim and was determined by Equation (6):

$$TR_{lim} = \frac{(ARL \times BW)}{(C \times CPS_o)} \tag{6}$$

where, *ARL* is maximum acceptable risk level (10-5, unitless); *BW* is average adult body in kg (60.7 kg for South African population); *C* is mean metal concentration in the different potato treatments; and $CPS_o$ is the cancer slope factor (6.3, 0.009 and 1.7 mg/kg/day for Cd, Pb and Ni, respectively) [53].

*2.5. Data Analysis*

All parameters (heavy metal analysis) and measurements (health risk assessment) were compared. All parameters and measurements were tested at $p < 0.05$ significance level and the Duncan multiple range test was used for separation between treatment means. Statistica v. 10, StatSoft was used for all statistical analysis.

## 3. Results

Some of the chosen elements, such V, Be, and Ba, were undetectable in all plant organs after receiving all fly ash acid mine drainage treatments. On the other hand, eleven other elements (Al, Pb, Co, Ni, Cd, Fe, Ca, Mg, Zn, Cu, and Mo) were detected in all parts of the potatoes that were irrigated with FA-treated AMD treatments (Table 1). The AMD treatments on potato tubers had a significant ($p \leq 0.01$) effect of reducing the concentration of Pb, Co, Ni, Cd, Fe, B, Zn, and Cu when compared to the control. According to Table 1, the treated acid mine drainage showed significant differences in the concentration of minerals such as $SO_4^{2-}$, Na, Ca, K, Mg, and Mo as they were higher.

There was a significant increase of lead (Table 1) in the ameliorated and untreated acid mine drainage. Also, 100% FA-AMD ratio demonstrated a significantly higher Pb concentration (0.02 mg/kg) in both cultivars compared to treatments that were ameliorated with FA and control (Table 1). Whereas control displayed significantly lower Pb concentration (0.01 mg/kg for *Fianna*) and (0.02 mg/kg *Lady rosetta*) cultivars compared to 75% FA/AMD with 0.00 mg/kg Pb for both cultivars respectively.

When irrigated with a control, the test potato tubers exhibited 1.88 mg/kg and 1.33 mg/kg Fe respectively for *Fianna* and *Lady rosetta* (Table 1). By contrast, 50% FA: AMD ratio showed 0.77 mg/kg and 0.65 mg/kg for both cultivars, respectively. It can be said therefore that potatoes irrigated with 100% AMD (untreated AMD) accumulated the highest Fe concentration (69.13 mg/kg and 67.18 mg/kg) compared to treatments that were ameliorated with FA and control (Table 1). This is an indication that the fly ash could have absorbed Fe ions thus reducing their availability in the potato crop.

The minerals (Table 1) translocated in the organ (tuber) of the potato crop. Nickel did not significantly affect both cultivars when irrigated with the control (tap water). The reading for *Fianna and Lady rosetta* recorded 0.09 mg/kg when irrigated with control. For 50% FA: AMD ratio, *Fianna* recorded 1.77 and 1.68 mg/kg for *Lady rosetta* respectively. While 75% FA: AMD ratio did not show a significant difference for both cultivars. Moreover, 100% AMD did not show any significant difference in the cultivar. Lastly, the treatment highly had an impact when cultivars were irrigated with FA: AMD ratios at $p \leq 0.001$.

Aluminum had no significant difference in the cultivars for control, for control *Fianna* and *Lady rosetta* both recorded 0.00 mg/kg. The same results were observed for 50% FA: AMD ratio. Whilst *Fianna* and *Lady rosetta* recorded 1.23 mg/kg when irrigated with 75% FA: AMD ratio. Lastly, when cultivars were irrigated with 100% AMD ratio *Fianna* recorded 5.39 ang 5.74 mg/kg for *Lady rosetta* respectively. Moreover, the treatment significantly had an impact when potato crops were irrigated with FA: AMD ratios at $p \leq 0.05$ and *Lady rosetta* recorded 69.13 and 67.18 mg/kg at $p \leq 0.001$.

**Table 1.** Heavy metal distribution from acid mine water drainage (AMD) water mixed with fly ash (FA) on the plant tissue (tuber) of two cultivars of potato. Values (Mean $\pm$ S.E.) followed by similar letters in a column are not significantly different at $p \leq 0.05$, ** $p \leq 0.01$, *** $p \leq 0.001$. NS = not significant.

| Organ | Ni | Cu | Fe | Zn | Al | Pb | Co | Cd |
|---|---|---|---|---|---|---|---|---|
| Tuber | | | | | | | | |
| Control—F | 0.09 ± 0.00 [b] | 0.00 ± 0.0 [a] | 1.88 ± 0.0 [b] | 0.19 ± 0.0 [a] | 0.00 ± 0.9 [ab] | 0.01 ± 0.0 [b] | 0.07 ± 0.0 [b] | 0.01 ± 0.0 [a] |
| Control—LR | 0.09 ± 0.00 [b] | 0.00 ± 0.0 [a] | 1.33 ± 0.0 [b] | 0.18 ± 0.0 [a] | 0.00 ± 0.9 [ab] | 0.02 ± 0.0 [b] | 0.06 ± 0.0 [b] | 0.07 ± 0.0 [a] |
| 50% FA: AMD—F | 1.77 ± 0.01 [c] | 0.12 ± 0.0 [a] | 0.77 ± 0.0 [a] | 0.15 ± 0.0 [a] | 0.00 ± 0.0 [a] | 0.00 ± 0.0 [a] | 0.16 ± 0.1 [a] | 0.98 ± 0.0 [b] |
| 50% FA: AMD—LR | 1.68 ± 0.00 [c] | 0.00 ± 0.0 [a] | 0.65 ± 0.0 [a] | 0.13 ± 0.0 [a] | 0.00 ± 0.0 [a] | 0.00 ± 0.0 [a] | 0.17 ± 0.1 [a] | 0.84 ± 0.0 [b] |
| 75% FA: AMD—F | 0.01 ± 0.00 [a] | 0.03 ± 0.0 [b] | 0.98 ± 0.1 [a] | 0.29 ± 0.0 [b] | 1.23 ± 0.00 [b] | 0.00 ± 0.0 [a] | 0.02 ± 0.0 [c] | 0.04 ± 0.0 [a] |
| 75% FA: AMD—LR | 0.01 ± 0.00 [a] | 0.02 ± 0.0 [b] | 0.50 ± 0.0 [a] | 0.22 ± 0.0 [b] | 1.23 ± 0.01 [b] | 0.00 ± 0.0 [a] | 0.01 ± 0.0 [c] | 0.03 ± 0.0 [a] |
| 100% AMD—F | 1.89 ± 0.00 [d] | 1.22 ± 0.0 [c] | 69.13 ± 0.4 [c] | 13.23 ± 0.0 [c] | 5.39 ± 0.3 [c] | 0.02 ± 0.0 [b] | 0.02 ± 0.0 [c] | 1.34 ± 0.0 [c] |
| 100% AMD—LR | 1.88 ± 0.01 [d] | 1.23 ± 0.0 [c] | 67.18 ± 0.2 [c] | 12.86 ± 0.0 [c] | 5.74 ± 0.02 [c] | 0.02 ± 0.0 [b] | 0.02 ± 0.0 [c] | 1.01 ± 0.0 [c] |
| F-Statistics | | | | | | | | |
| Organ | 14.58 *** | 9.00 ** | 6.65 *** | | 3.22 * | 20.65 ** | 5.70 ** | 4.35 ** |

* F—*Fianna* (cultivar 1); * LR—*Lady rosetta* (cultivar 2); * Control (tap water); * 50% FA: AMD; * 75% FA: AMD; * 100% AMD; a–d Means within a single variable and column followed by a different letter are significantly different according to the Fisher's least significant difference test (0.05, 0.01 and 0.001 level).

The results of the current study revealed a significant difference in the concentration of Cu in tissues of potatoes grown with the application of ameliorated and untreated acid mine drainage (Table 1). Especially, irrigation with 100% AMD (untreated AMD) recorded a Cu concentration of 1.22 mg/kg and 1.23 mg/kg for the respective cultivars compared to treatments that were ameliorated with FA and control (Table 1). On the other hand, potatoes supplied with a control displayed significantly lower Cu: 0.00 mg/kg for the *Fianna* and

0.00 mg/kg for *Lady rosetta* cultivars compared to watering with 75% FA: AMD ratio which resulted in 0.03 mg/kg Cu in *Fianna* and 0.02 mg/kg in *Lady rosetta*. All FA-treated AMD irrigated potatoes had significantly higher Cu levels compared to the control (Table 1). These figures met the CODEX general standard limits for food contaminants and toxins in food and feed (CODEX STAN 193-1995) which recommends a maximum 0.3 mg/kg for potato. The WHO and FAO recommends that the concentration of Cu in food should not be above 30 mg/kg while that of the NAFDAC is 20 mg/kg for food. Table 1 shows that the fly ash was able to ameliorate the copper ions in the 100% AMD ant thus rendering the potatoes safe to consume.

On the other hand, untreated acid mine drainage recorded the highest Ni as *Fianna* had 1.89 mg/kg, while *Lady rosetta* had a 1.88 mg/kg. It should be noted that there are no specified standard limits for Ni for metals in foods and vegetables FAO/WHO, NAFDAC nor in CODEX standards.

The concentration of Cd in stems (Figure 2) of potatoes irrigated with acid mine drainage was relatively higher (Table 1). For example, irrigation with a control recorded 0.08 mg/kg Cd in *Fianna* and 0.07 mg/kg Cd in *Lady rosetta*. When untreated AMD was used to water the test potatoes, it elicited significantly high Cd concentration: 2.01 mg/kg for *Fianna* and 4.61 mg/kg for *Lady rosetta* compared to treatments that were ameliorated with FA and control. On the other hand, the application of 50% FA-AMD displayed significantly lower Cd concentration of 1.01 mg/kg and 0.88 mg/kg for *Fianna* and *Lady rosetta* cultivars, respectively.

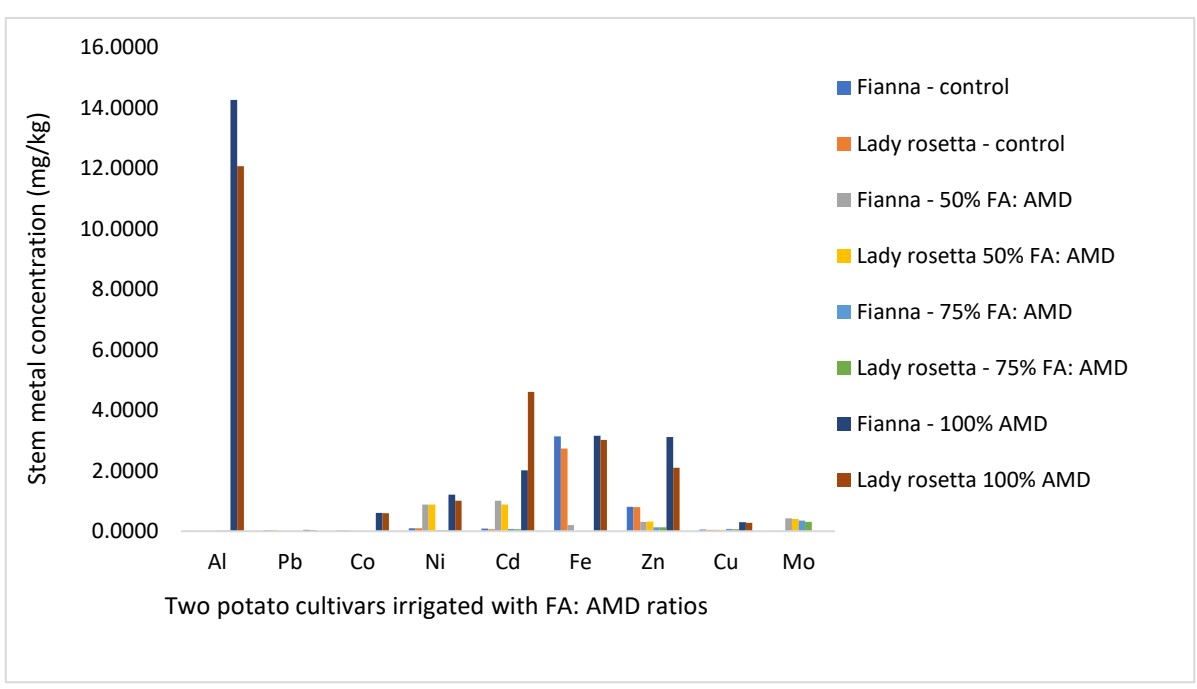

**Figure 2.** Concentration of heavy metals in the stem during growth of *Fianna* and *Lady rosetta* cultivar irrigated with acid mine drainage.

The ratio of the studied elements' concentration in leaves to tubers was consistently higher in *Fianna* than in *Lady rosetta*. This support the hypothesis that differences in final tuber concentration between cultivars are related to element loading into the phloem or element movement once loaded into the phloem. The mechanisms driving the re-distribution of mineral elements from leaves to tubers, nonetheless, may not always be related to those governing Cd transport. Mineral element concentrations in the leaves of cultivar *Fianna* were comparable to or higher than those of cultivar *Lady rosetta*. The only element with higher quantities in *Fianna*'s leaves was cadmium.

Recurrent irrigation of the potato cultivars with FA-AMD radically affected new and old leaves (Figure 3). The concentration of Fe in old leaves was 2.22 mg/kg and 2.02 mg/kg respectively for *Fianna* and *Lady rosetta* when irrigated with a control. In contrast, the Fe concentration in new leaves was 0.66 mg/kg and 0.58 mg/kg for *Fianna* and *Lady rosetta*, respectively. As to be expected, Fe was high in the untreated acid mine drainage in new leaves (1.29 and 1.23 mg/kg for *Fianna* and *Lady rosetta*, respectively) and old leaves (3.54 mg/kg for *Fianna* and 3.62 mg/kg for *Lady rosetta*).

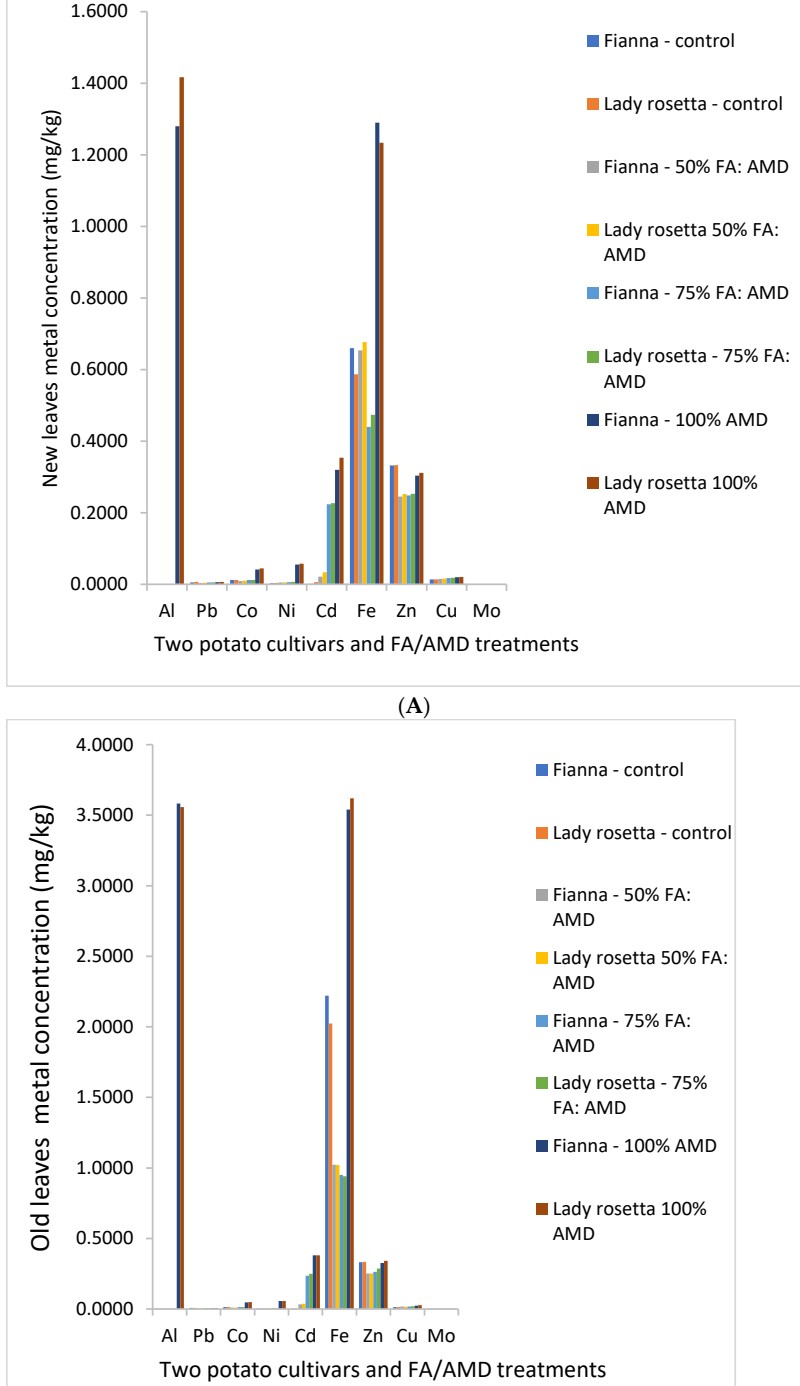

(**A**)

(**B**)

**Figure 3.** Concentration of heavy metals in the new leaves (**A**) and old leaves (**B**) during growth of *Fianna* and *Lady rosetta* cultivar irrigated with acid mine drainage.

With regards to Cd, irrigating with a control showed a concentration of 0.00 mg/kg for both cultivars in older leaves. Interestingly, new leaves had no Cd content. When the crops were irrigated with untreated acid mine drainage, they showed 0.32 mg/kg and 0.35 mg/kg for *Fianna* and *Lady rosetta*. Nonetheless, untreated acid mine drainage proved to account for significantly increased Ni concentration in both the older and new leaves of both cultivars 0.05 mg/kg and 0.05 mg/kg for both cultivars.

The soil-plant transfer coefficient (Figure 4) of *Fianna* and *Lady rosetta* cultivars irrigated with untreated and FA: AMD ratios. There is a significance difference between cultivars when the crops are irrigated with the 0% acid mine drainage (control) Cd recorded 2.5 and 14.96 for *Fianna* and *Lady rosetta*, respectively. For the rest of the treatments 50% FA: AMD, 75% FA: AMD ratio, and 100% AMD (untreated AMD), *Fianna* showed higher soil-plant transfer coefficient percentages. The same pattern is observed for Pb, *Lady rosetta* showed higher percentages in the control with values of 191.80 and 246.66 for *Fianna* and *Lady rosetta*, respectively. Moreover, *Fianna* recorded higher percentages in three other treatments as observed with Cd. For Cu, 100% AMD ratio (untreated AMD) recorded higher soil-plant transfer coefficient compared to other treatments. Cultivars did not show a significance difference when irrigated with untreated acid mine water, *Fianna* recorded 6.15 whilst *Lady rosetta* recorded 6.18.

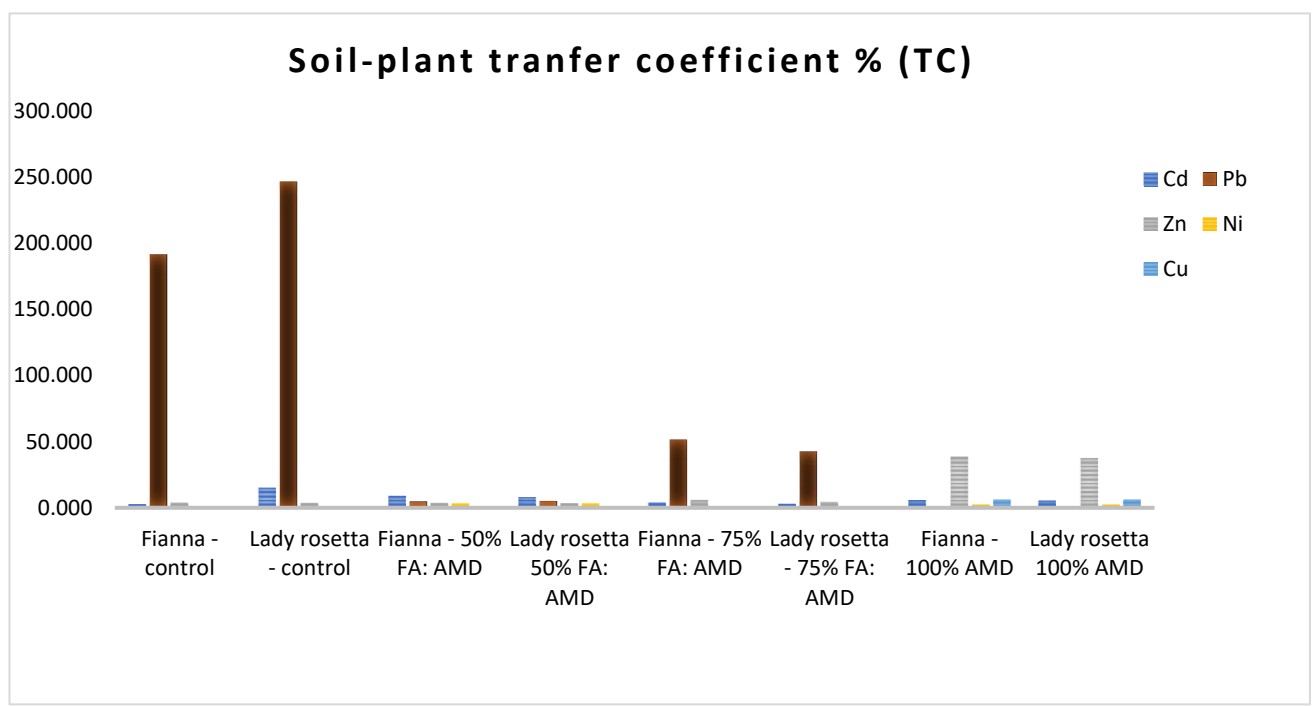

**Figure 4.** Soil-Plant Transfer Coefficient % (TC) of *Fianna* and *Lady rosetta* cultivar irrigated with acid mine drainage water. Values (Mean ± S.E).

Results (Table 2) showed a high significant of bioaccumulation of metals in the potato samples. For example, when the *Fianna* was irrigated with 75% AMD, it recorded 11.83 mg/kg Cd while its counterpart exhibited 12.50 mg/kg Cd. Furthermore, when both cultivars were irrigated with untreated acid mine drainage, they had 2.37 Cd. The concentration of Zn, Ni and Cu were similar between the test potato cultivars and treatment thereof. This can be explained by the high concentration of Cd found in the soil and water samples we have analyzed in this study.

**Table 2.** Effect of AMD treatments on the bioaccumulation factor (BF) of two potato varieties (*Fianna* and *Lady rosetta*).

| | | Bioaccumulation Factor (BF) | | | | |
|---|---|---|---|---|---|---|
| Treatments | Cultivars | Cd | Pb | Zn | Ni | Cu |
| Control | *Fianna* | 0.23 | 0.36 | 0.45 | 0.16 | 0.23 |
| Control | *Lady rosetta* | 0.20 | 0.38 | 0.45 | 0.17 | 0.23 |
| 50% FA: AMD | *Fianna* | 0.78 | 1.17 | 0.03 | 0.00 | 0.15 |
| 50% FA: AMD | *Lady rosetta* | 0.86 | 1.10 | 0.03 | 0.00 | 0.14 |
| 75% FA: AMD | *Fianna* | 11.83 | 5.43 | 0.19 | 0.00 | 0.01 |
| 75% FA: AMD | *Lady rosetta* | 12.50 | 5.62 | 0.21 | 0.00 | 0.01 |
| 100% AMD | *Fianna* | 2.37 | 0.02 | 0.00 | 0.00 | 0.04 |
| 100% AMD | *Lady rosetta* | 2.37 | 0.02 | 0.00 | 0.00 | 0.04 |

The average daily intake (EDI) of Pb, Cd, Zn, Ni and Cu were estimated according to the mean concentration of each metal in each potato cultivar in all treatments. The EDI of studies metals from consumption of potato samples are shown in Table 3. The recorded average figures of EDI for Cd, Pb, Zn, Ni and Cu in the *Fianna* cultivar for control were 3.43, 5.13, 5.15, 2.45 and 1.07 (mg/person/day). With regards to the other cultivar (*Lady rosetta*), the following readings were recorded: 1.96, 6.60, 4.86, and 4.46 (mg/person/day) respectively. It is evident that the EDI differed significantly in all treatments.

**Table 3.** Effect of AMD treatments on the Translocation factor (TF) and Estimated Daily Intake (EDI) of two potato varieties (*Fianna* and *Lady rosetta*).

| | | Translocation Factor (TF) | | | | Estimated Daily Intake (EDI) | | | |
|---|---|---|---|---|---|---|---|---|---|
| Treatments | Cultivars | Cd | Pb | Zn | Ni | Cd | Pb | Zn | Ni |
| Control | *Fianna* | 0.30 | 0.37 | 1.72 | 0.03 | 3.48 | 5.13 | 5.15 | 2.45 |
| Control | *Lady rosetta* | 1.46 | 0.31 | 1.83 | 0.03 | 1.96 | 6.60 | 4.86 | 2.48 |
| 50% FA: AMD | *Fianna* | 17.74 | 0.87 | 1.67 | 0.00 | 2.62 | 1.31 | 4.02 | 4.74 |
| 50% FA: AMD | *Lady rosetta* | 16.70 | 0.88 | 1.82 | 0.00 | 2.25 | 1.34 | 3.70 | 4.52 |
| 75% FA: AMD | *Fianna* | 23.66 | 0.78 | 0.88 | 0.33 | 1.07 | 1.84 | 7.99 | 5.11 |
| 75% FA: AMD | *Lady rosetta* | 18.75 | 1.01 | 1.29 | 0.36 | 8.03 | 1.48 | 5.90 | 4.95 |
| 100% AMD | *Fianna* | 19.94 | 0.30 | 0.02 | 0.03 | 3.59 | 5.69 | 3.54 | 5.06 |
| 100% AMD | *Lady rosetta* | 15.09 | 0.32 | 0.00 | 0.03 | 2.72 | 5.39 | 3.43 | 5.03 |

Threshold hazard quotient (THQ) values for the individual heavy metals through the consumption of potatoes are shown in Table 4. The THQ values in Pb, Zn, Ni and Cu were evaluated based on available slope factors. The cancer risk (Table 4) for all three metals (Pb, Zn and Ni). The 50% FA: AMD ratio recorded 0.480 and 0.413 for *Fianna* and *Lady rosetta* cultivars, respectively. Nonetheless, treatment untreated AMD recorded the highest with values (0.668 and 5.15) for both *Fianna* and *Lady rosetta* respectively. The recorded average figures of potential carcinogenic effects for Cd, Pb and Ni in the *Fianna* cultivar for all treatment were 0.00 mg/kg (Table 5). Nickel reported 0.00 mg/kg in all treatments except in 75% FA: AMD ratio which recorded 0.02 mg/kg in both cultivars.

It can be concluded from the analysis that irrigating the selected potato cultivars with untreated acid mine drainage caused the accumulation of more heavy metals when compared with the other three treatments. Overall, the threshold hazard quotient (THQ) values were >1 for all heavy metals and it can be concluded that if the potatoes subjected to these treatments were consumed, they could not pose health risks.

**Table 4.** Effect of AMD treatments on the Target Hazard Quotient (THQ), Target cancer risk (TR) and Cancer risk of two potato varieties (*Fianna* and *Lady rosetta*).

| Treatments | Cultivars | Target Hazard Quotient (THQ) | | | | Target Cancer Risk (TR) | | | | Cancer Risk |
| | | Cd | Pb | Zn | Ni | Cd | Pb | Zn | Ni | |
|---|---|---|---|---|---|---|---|---|---|---|
| Control | *Fianna* | 0.00 | 0.00 | 0.00 | 0.00 | 0.01 | 0.00 | 0.00 | 0.00 | 0.01 |
| Control | *Lady rosetta* | 0.03 | 0.00 | 0.00 | 0.00 | 0.03 | 0.00 | 0.00 | 0.00 | 0.04 |
| 50% FA: AMD | *Fianna* | 0.45 | 0.00 | 0.00 | 0.02 | 0.45 | 0.00 | 0.00 | 0.02 | 0.50 |
| 50% FA: AMD | *Lady rosetta* | 0.39 | 0.00 | 0.00 | 0.01 | 0.39 | 0.00 | 0.00 | 0.02 | 0.41 |
| 75% FA: AMD | *Fianna* | 0.01 | 0.00 | 0.00 | 0.00 | 0.02 | 0.00 | 0.00 | 0.00 | 0.20 |
| 75% FA: AMD | *Lady rosetta* | 0.01 | 0.00 | 0.00 | 0.00 | 0.01 | 0.00 | 0.00 | 0.00 | 0.01 |
| 100% AMD | *Fianna* | 0.62 | 0.00 | 0.01 | 0.02 | 0.63 | 0.00 | 0.01 | 0.02 | 0.70 |
| 100% AMD | *Lady rosetta* | 0.47 | 0.00 | 0.01 | 0.02 | 0.50 | 0.00 | 0.01 | 0.02 | 0.51 |

**Table 5.** Effect of AMD treatments on the Potential carcinogenic effects of two potato varieties (*Fianna* and *Lady rosetta*).

| Treatments | Cultivars | Potential Carcinogenic Effects | | |
| | | Cd | Pb | Ni |
|---|---|---|---|---|
| Control | *Fianna* | 0.00 | 3.71 | 0.00 |
| Control | *Lady rosetta* | 0.00 | 2.89 | 0.00 |
| 50% FA: AMD | *Fianna* | 0.00 | 14.51 | 0.00 |
| 50% FA: AMD | *Lady rosetta* | 0.00 | 14.26 | 0.00 |
| 75% FA: AMD | *Fianna* | 0.00 | 10.36 | 0.02 |
| 75% FA: AMD | *Lady rosetta* | 0.00 | 12.88 | 0.02 |
| 100% AMD | *Fianna* | 0.00 | 3.35 | 0.00 |
| 100% AMD | *Lady rosetta* | 0.00 | 3.54 | 0.00 |

## 4. Discussion

Nickel (Ni) is needed by plants for the absorption of iron and seed germination however its deficiency causes plants to produce nonviable seeds [51–53]. Its application through fertilizer protects crops from certain yield-limiting diseases, potentially lowering pesticide use and increasing crop production [54–56]. Irrigating the *Fianna* and *Lady rosetta* cultivars with a control resulted in similar concentration of Ni in their tissues (0.09 mg/kg). The fly ash that was mixed with AMD seemed to have ameliorated the acid mine drainage hence the tubers recorded the lowest amount of Ni. This was shown when the potatoes were irrigated with the 75% FA: AMD ratio, which resulted in accumulation of 0.01 mg/kg Ni in the *Fianna* and 0.19 mg/kg Ni in *Lady rosetta*. Results in Figure 2 are supported that by [57–59] where fly ash was said to have adsorbed Ni ions in acid mine drainage water.

Other studies [60,61] also observed a higher Cd content in older leaves of potato. The results of this study as shown in Figure 3A,B illustrates that the Ni content in both older and new leaves was not traceable particularly when plants were irrigated with a control. Similarly, the irrigation with 50% FA: AMD showed no traces of Ni as none was recorded for both older and new leaves of both cultivars. The above results are support that by [62] where they found no Ni in both old and new leaves on both cultivars. This observed growth inhibition in both cultivars could be attributed possibly to phyto-accumulation of heavy metals in plant organs. Heavy metal stress is reported to have a suppressive effect on the overall growth of plants, which for example can be manifested in reduced shoot growth in potato [63–65] and spinach [66], and a decline in the number of pods and seed yield in pigeon pea [67–69].

On the other hand, watering with 50% FA: AMD recorded 1.77 mg/kg and 1.68 mg/kg Ni for *Fianna* and *Lady rosetta* cultivars, respectively. Some of the selected elements such

as V, Be and Ba were below detection in all plant organs in all fly ash acid mine drainage treatments. This is because these elements are adsorbed and are not readily available [70].

Knowledge on the elemental concentrations in irrigation water is important and in cases where they are relatively low compared to that recommended for irrigation water in South Africa [71] however, in the long-term, irrigating plants with these treatments can lead to a build-up of metals in soils, their absorption, translocation, and accumulation in plant organs. Also, it is crucial to know the concentration of heavy metals in soils in croplands especially soils used in experimental settings. For this study, selected chemical properties of the soil were analyzed, particularly to assess if it contained toxic metals. Results of this study show a significant effect of irrigating the test potatoes with AMD mixed with FA on the translocation of heavy metals in the tissues of potatoes. Nonetheless, the selected cultivars revealed no significant difference in both seasons for both cultivars when irrigated with all four FA: AMD ratios.

The differences were probably due to genetic differences in their inherent tuber sizes that is, the *Fianna* cultivar generally has a bigger tuber size compared to *Lady rosetta*. In their study, reported that tubers from different cultivars tend to exhibit varying sizes and weights, propably as a result of genetic make-up [72]. The results of this study support that by [73] who reported that different irrigation regimes affect cultivars differently due to diverse genotypes. The explanation for the difference noted in the 75% FA: AMD ratio could be that it contained fewer toxins and therefore the fly ash could have been able to adsorb the heavy metals in the acid mine drainage. Such results agree with that earlier shown by [74,75]. A study by Bilate and Mulualem (2016) [76] showed that heavy metal concentration in irrigation water significantly reduces the tuber fresh and dry weight of *Fianna* and *Lady rosetta* cultivars. Their results were in agreement with previous findings by [77], where they agreed that the acid mine effluents had considerably affected the tuber growth and the size thereof.

Potato plants are known to flower and start tuberization 8 weeks after planting, a period during which major and trace elements are needed. In this study, the required macro-micronutrients could have been supplied by the AMD and Fly Ash. Invariably, the fact that the cultivars responded differently in terms of growth and tuber yield to long-term exposure to the treatments could suggest genetic differences in their ability to absorb, translocate and accumulate heavy metals. These findings in this regard are in agreement with studies by [78,79]; who reported genetic differences in the accumulation of Cd and other metals in organs of two potato cultivars cultivated in heavy metal-contaminated soils in Australia.

Literature background has reported on distribution of Cd between tissues of matured potato plants given that the concentration of Cd in matured plants has implications to human health. Cadmium is critical because it has major health consequences; renal impairment caused by chronic dietary Cd exposure is characterized by proximal tubule dysfunction [80]. For this study, regular irrigation of potato with a control ratio drastically increased the Cd content in tubers (0.01 mg/kg and 0.07 mg/kg for *Fianna* and *Lady rosetta* respectively. The concentration of Cd was above the WHO acceptable limits in untreated acid mine drainage. Table 1 showed that *Fianna* recorded 1.34 mg/kg Cd whilst that in *Lady rosetta* was slightly lower (1.01 mg/kg). Nonetheless, FA was able to significantly reduce the Cd ions in both FA-AMD treatments (Figure 2): Cd content was 0.98 mg/kg and 0.84 mg/kg in 50% FA: AMD ratio for *Fianna* and *Lady rosetta* respectively, while the recommended maximum limit is 0.1 mg/kg [81]. The readings were slightly high according to the CODEX general standard for food contaminants and toxins in food and feed (CODEX STAN 193-1995). On the other hand, in 75% FA: AMD ratio the Cd concentration was 0.04 mg/kg and 0.03 mg/kg for *Fianna* and *Lady rosetta* individually. It is thus evident that FA adsorbed the Cd ions and both cultivars meet the CODEX standard, therefore the potato crop irrigated with a 75% FA: AMD ratio ratio can be consumed by humans without causing any detrimental effects.

According to [82–84] Cd intake via the food chain poses a major hazard to human health because it is potentially implicated in cancer, kidney tubule damage, and bone fractures. The finding of this study therefore is significant in that Cd can be reduced to acceptable levels by 75% FA: AMD ratio. Mixing FA with AMD at a ratio of 75% FA: AMD has also been shown by [85–90] to adsorb the heavy metals. Other studies [90–92] attested that despite that Cd is a non-essential element, its ions are distributed throughout organs of potato cultivars, and it is detrimental to plant growth and human health. The aim of this study was to see if differences in tuber heavy metal concentrations were due to inherently increased plant uptake by cultivar *Fianna* or changes in heavy metal dispersion throughout the plant. In the main, the concentration of mineral nutrients in *Fianna* tubers was higher than in *Lady rosetta* tubers. The concentration of Pb was significantly higher in all FA-amended AMD treatments including 100% AMD compared with the control (Table 1). The values of Pb shown in this study meet the maximum allowable limit for food and feed which is 0.1 mg/kg (CODEX STAN 193-1995). In conclusion, all the FA-AMD treated potatoes had Pb concentration that was below the minimum standard limits, and this proves that FA was able to adsorb the Pb ions in the tuber samples. Importantly, it reduces the metal's concentration, a finding also reported by [90–92].

Supplying 75% FA: AMD significantly decreased the Fe concentration in the tubers as they recorded 0.98 mg/kg and 0.50 mg/kg for both cultivars, respectively. This means that the 75% FA: AMD ratio was able to reduce the element's concentration drastically. The above results support that earlier shown by [93,94] and confirm the ability of fly ash to absorb the Fe ions especially from the 100% AMD (untreated AMD) ratio. The maximum limit of FAO/WHO guidelines for metals in foods and vegetables for Fe is 48 mg/kg. Given the results shown in Table 2, it is evident that irrigating the test cultivars with the 75% FA: AMD does not elicit increased Fe and the tubers can be consumed and not cause any significant detrimental effects on the health of humans. It should be noted however that the application of Fe at lower and higher rates than those suggested are deemed inefficient because they are thought to limit plant development, decrease chlorophyll content, and inhibit photosynthesis [95,96].

Overall, older leaves of both cultivars exhibited greater Fe compared to new leaves (Figure 3A,B). This could likely be because older leaves are likely to have accumulated Fe over time as described by [97].

The importance of micronutrients in health and nutrition cannot be overstated, and zinc is one of them [98,99]. In fact, the value of Zn is increasingly recognized especially in human health because its shortage especially in staple diets may play a significant role in the onset of diseases [100,101]. Analysis done in this research show that tubers irrigated with tap water and untreated acid mine drainage exhibited significant increased Zn concentration (Table 1). Potato tubers irrigated with tap water recorded Zn concentrations of 0.19 mg/kg and 0.18 mg/kg for *Fianna* and *Lady rosetta* cultivars, respectively. On the other hand, when the *Fianna* cultivar was watered with the 50% FA: AMD ratio, it showed 0.15 mg/kg while its counterpart had 0.13 mg/kg Zn. These values were relatively lower compared to potatoes irrigated with untreated acid mine drainage. It is evident that the fly ash was able to ameliorate Zn ions in the acid mine drainage water, nonetheless, when irrigated with acid mine drainage, *Fianna* exhibited 13.23 mg/kg Zn while *Lady rosetta* showed 12.83 mg/kg. Overall, the concentration of Zn was increased in potatoes that were not treated with AMD. Lastly, the concentration of Zn in all the test treatments was below that recommended in the FAO/WHO guidelines for metals in foods and vegetables which are 60 mg/kg, 50 mg/kg in the NAFDAC and CODEX standards.

The EDI values shown in Table 2 proved that Cd is consumed above the permitted maximum tolerable daily intake however, Pb, Zn, Ni and Cu were below the permitted values which were obtained from the World Health Report [73]. This compared to plants irrigated with 75% FA: AMD ratio which revealed Cd concentration of 0.08 mg/kg for the *Fianna* and 0.07 mg/kg for *Lady rosetta*. In this case, the differences could probably be attributable to genotypic variations in the partitioning of Cd between the tubers and the

rest of the plant, particularly the proportion of Cd that was retained in the roots and leaves. This argument was recently advanced by [102,103], that genotypic variations in terms of heavy metals exist in plants e.g., in sweet potatoes and tomatoes. It was proven in this study that there was no significant net influx of metals into the shoot tissue as none were found in the shoot. This could be due to a disruption in Cd import into the leaf tissue or a balance between Cd transport into the leaf tissue and re-translocation to other plant tissues as was also proposed by [104] on different seasonal crops mustard (*Brassica campestris*), cauliflower (*Brassica oleracea* var. botrytis), cabbage (*Brassica oleracea* var. capitata), and spinach (*Brassica oleracea* var. capitata).

Considering xylem flow controls, it is probable that Cd is imported into leaf tissue, and it is unlikely that it will stop being carried to the leaves because xylem flow is required to replace water lost during transpiration [105]. The translocation and mobility of heavy metals have been researched in plants such as beans and wheat. For example, studied the absorption, transport, and mobility of radionuclides applied to leaves of beans [106]. These radioactive isotopes had varied mobility in the phloem, according to the researchers. Confirmed that phloem transport provided the metal micronutrients Mn, Zn, Co, and Fe and that this transport was linked to the availability of nicotianamine [107]. In addition to its role in long-distance transport, found that nicotianamine is critical for the regulation of metal transfer within cells [108]. According to [75], a threshold hazard quotient that is less than 1 mean that the vegetable can be consumed, and it cannot cause detrimental effect on the human health.

The most plausible method for Cd transport to tubers appears to be from the soil through basal roots to xylem shoots, followed by re-translocation to tubers in the phloem [109]. The difference in mineral nutrient contents in the tubers of the two varieties used in this study was unexpected. This may suggest that there are fundamental differences in the phloem loading mechanisms of the two cultivars as was also postulated by [110] where the concept of differential phloem loading systems was suggested within the same plant species. The ratio of an element's concentration in the leaves to that in the tubers can be used to estimate its phloem mobility [35]. A higher ratio implies that a bigger percentage of that element is still in the leaves and is not being transported to the tuber via the phloem [111].

## 5. Conclusions

In summary, the concentration of Cd was over the WHO acceptable standards in untreated acid mine drainage water. *Fianna* recorded 1.34 mg/kg while a notably decrease was observed in *Lady rosetta* with (1.01 mg/kg). In any case, FA was proven to significantly lessen the Cd particles in both FA-AMD ratios: Cd content was 0.98 mg/kg and 0.84 mg/kg in 50% FA: AMD ratio for *Fianna* and *Lady rosetta* separately, while the recommended maximum limit is 0.1 mg/kg. With the readings recorded, they are slightly high according to the CODEX general standard for food contaminants and toxins in food and feed (CODEX STAN 193-1995). Then again, in 75% FA: AMD ratio, the Cd concentration was 0.04 mg/kg and 0.03 mg/kg for *Fianna* and *Lady rosetta* exclusively. It is in this way obvious that FA adsorbed the Cd ions, and the two cultivars fulfill the CODEX guideline, nonetheless the potato crop irrigated with a 75% FA: AMD ratio ratio can be consumed by humans without causing any detrimental effects. It was the main purpose of this project to describe the basic features of Cd, Zn, Pb, and Ni uptake and translocation, and to uncover the reasons behind genotypic differences in heavy metal accumulation in tubers. Lastly, according to literature a threshold hazard quotient that is less than 1 mean that the vegetable can be consumed, and it cannot cause detrimental effect on the human health. Lead proved to have the highest potential carcinogenic effect and value slightly higher which were obtained from United State Environmental Protection Agency.

**Author Contributions:** M.V.R. assisted with the experimental design as well as data collection. N.I.M. and R.M. assisted with data analysis and manuscript formulation of the manuscript entitled "Heavy Metal Risk Assessment of Potato (*Solanum tuberosum* L.) Cultivars irrigated with Fly Ash-Treated Acid Mine Drainage". All authors have read and agreed to the published version of the manuscript.

**Funding:** This research received no external funding.

**Data Availability Statement:** Not Applicable.

**Acknowledgments:** Not Applicable.

**Conflicts of Interest:** The authors declare no conflict of interest.

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
