# Peer review of "Heavy Metal Analysis and Health Risk Assessment of Potato (Solanum tuberosum L.) Cultivars irrigated with Fly Ash-Treated Acid Mine Drainage"

_horticulturae, doi:10.3390/horticulturae9020192_

Round 1
Reviewer 1 Report
horticulturae-2158305-review
The study assesses the levels of heavy metals, and health risk assessment in two potato crop cultivars, namely, Fianna and Lady Rosetta (both determinate and indeterminate) when exposed to irrigation with different fly ash: acid mine drainage amelioration ratios. The study assesses the health risk assessment in the potato tissues (stem, tubers, new and old leaves) of the potato cultivars. This is an interesting and meaningful research topic. This paper has a detailed and rigorous experimental design. Some suggestions are as follows:
1. Some pictures from the experiment should be added to help readers understand. For example, photos of experimental materials and sample preparation process.
2. Some figures are non-standard, such as Figure 1 and Figure 4. One decimal place is recommended for longitudinal axis.
3. The authors cite a large number of references, and should carefully check whether the manuscript corresponds to the front and back, according to the publication format of the Journal. Some works maybe helpful, such as: https://doi.org/10.1007/s10230-020-00701-x, https://doi.org/10.1007/s10230-022-00886-3, https://doi.org/10.3390/w14244093.
4. Lines 213, [38-39] reported that metals translocate from soil and to plant parts (root and shoot). Who reported, missing subject.
5. Lines 213, equation (2); Lines 228, Eq (3), It should be consistent.
Author Response
The authors have addressed all comments.

Reviewer 2 Report
The manuscript reported translocation features of heavy metals in two potato cultivars irrigated by acid mine drainage treated with fly ash. The greenhouse experiment was well-designed, and the data were adequate. The writing was nice overall. However, the data presentation, statistics, and visualization need to improve. All tables and figures are supposed to reorganize (some were missing). Besides, the article could be more concise.
My major concern focuses on statistics. According to the experiment design, the author would like to reveal the effects of FA treatment, cultivar, and their interaction on heavy metal uptakes. It requires a two-way ANOVA. The authors said they did it (L284), but I did not find their results. Note that the control group (tap water) should be removed in this two-way ANOVA. This could be the main evidence for the conclusions of this article.
Another problem is the treatment naming. There were so many names for the same treatment, e.g., treatment 3:1, 3:1 FA, 3:1, 3:1 FA-AMD. 3:1 FA:AMD, 75% FA:AMD, 75% AMD, T3, Treatment 3, etc. It is hard to follow. The cultivar names are also confusing. I am still unsure whether 3:1 contained 75% AMD and 25% FA. Is the purification of 25% FA better than 50% FA? Why?
Minor concerns:
(1) L15, the specific treatments should be explained here. The abstract should be self-explanatory.
(2) L23, where does the recommended maximum limit come from? Which standard?
(3) L25, treatment 3:1 is hard to follow for those who only read the abstract.
(4) The authors are supposed to describe the results of all heavy metals, not only Cd.
(5) L78 and L92, remove these subtitles.
(6) L126, the mechanisms of fly ash purifying AMD should be introduced.
(7) omit “such as the authors of”.
(8) The effect of genetic variation of cultivars on heavy metal uptakes should be involved in the section Introduction since I saw some discussions on this in the section Discussion. The differences between two cultivars can be introduced in detail (L172). What does determinate mean? However, the effect should be identified by a two-way ANOVA, as recommended above.
(9) L177, omit (5).
(10) How many pots or plants per block?
(11) L190, check the degree symbol.
(12) L278, which R version? What’s lsmeans?
(13) L286, keep space around <. Keep P in italics and capital. Check throughout.
(14) L291, where Table 1?
(15) L291, where season 2? Were all results consistent between the two seasons?
(16) L293, the letters must come from multiple comparisons at the same level (usually 0.05, as L300).
(17) L294, the letter a usually denotes the largest number.
(18) L296-299, it is hard to follow. Each table should be self-explanatory.
(19) L301, Fisher’s test was not mentioned in MM.
(20) L314, Figure 2 didn’t show much information. It could be removed since Table 2 tells the same thing.
(21) Never cite references in the section Results!!! The results are too long. Just list the results of this study.
(22) L366, hard to distinguish the points in Figure 2 as well as Figure 3. Vibrate these points on the x-axis. No lines were needed in Figure 3A.
(23) L366, does the x axis mean?
(24) L366, omit (a).
(25) L367, omit Values …
(26) L397-398, might the tap water contain Cd?
(27) L435, Figure 4 is hard to understand.
(29) The main text, excluding tables and figures, should also be self-explanatory, so never start a sentence with “Figure 3 show(ed)” (L317), “as shown in Figure 3” (L401), “Figure 4 depicts” (L438), or “Table 5 results showed” (L462).
(30) L482, the sentence is not proper here. Omit it.
(31) Table 5? Where were Tables 3 and 4?
(32) Table 5 could be divided into several tables. Remember to use three-line tables.
(33) L501-506, so did the authors measure the heavy metals of AMD, treated-AMD, tap water, and soils cultivating plants?
(34) L627, the section Conclusion should be shorter. Some sentences (e.g., L649-664) could be moved to the section Discussion or MM.
Author Response

(The authors gave the same response as above.)

Reviewer 3 Report
The study evaluates levels of heavy metals in potato cultivars irrigated with ADM and the role of FA in attenuating these metals in the edible material (potatoes) and risks to human health. This theme is little researched in Africa and is of great importance for food production in the region with a minimum of security for the health of the population. However, an in-depth review of the manuscript is required to have minimum conditions for publication. At the end of the introduction, it is indicated that ADM has high concentrations of heavy metals, but it is necessary to indicate which chemical elements exceed levels considered toxic in the ADM solution. The authors need to separate heavy metals that are potentially toxic to humans and heavy metals considered plant nutrients that do not pose a risk to human health, such as Fe. Therefore, not all the chemical elements present in ADM are a risk to human health and this needs to be very clear in the introduction. The manuscript does not make clear the maximum critical levels of heavy metals present in irrigation water to be used in agriculture. Brazil established these critical levels by the institution CONAMA (Brasil, 2005). Below is the reference and the link (below) with these critical values of heavy metals contained in irrigation water when using wastewater from industry or from other places such as ADM. Brazil. National Council for the Environment. 2005. CONAMA Resolution No. 357, of March 17, 2005. Provides for the classification of bodies of water and environmental guidelines for their classification, as well as establishes the conditions and standards for the release of effluents, and other measures. Brasilia: Official Gazette of the Union. http://www.siam.mg.gov.br/sla/download.pdf?idNorma=2747 This information is important so that the researcher can study only the chemical element present in the ADM that really has the greatest risk of contamination for the plant. This decision would avoid evaluating chemical elements without potential risk and this would allow further studies evaluating the elements in the soil and in the leaf, that is, the potential of ADM to contaminate the soil and impair the nutritional status of the plant and, consequently, the productivity and still generate edible product contaminated. The authors indicated an average temperature in the greenhouse, but it is important to change it to maximum (average) and minimum (average) temperature of the air. Describe in detail the irrigation management before and after the imposition of treatments. What was the maximum water retention capacity of the soil adopted in the experiment? What method was used to apply water to the soil? Did the pot have a hole in its base? Was there a percolated water collector in the soil at the bottom of the pot? Another important point in the manuscript refers to the decontamination of potato samples that were used for chemical analysis. There is a great risk that the results obtained regarding the levels of heavy metals in the potato came from the soil and not from the plant tissue (potato). It is hardly possible to decontaminate the potato from soil residues. Therefore, method of washing potatoes must be detailed. Why was analysis of fresh plant material performed? These results are very variable depending on the water content in the potato. Why was the content of heavy metals in the soil not evaluated? It remains to improve the discussion by avoiding just a description of the results, but rather the mechanisms involved that explain higher levels of certain chemical elements present in the analyzed plant material (potato). It remains to indicate work with Cd in potato plants and guide the defense mechanisms of this species to nutritional stress due to Cd toxicity. This needs to be further discussed.
Author Response

(The authors gave the same response as above.)

Round 2
Reviewer 2 Report
The authors addressed all of my minor concerns, but not my major one.
It is a pity that the authors removed all conclusions on the effects of variables. I still can't find the results of the two-way ANOVA, which the authors stated in L299 of the current version (remove it if the authors resist performing this analysis). The paper loses some scientific significance and remains the descriptive results of heavy metal uptake under different conditions without mechanism analysis.
However, the current conclusions are well supported by the current data presentation. It may reach the scope and level of this journal. I have no further suggestions if the authors insist on publishing the current version.
Author Response
The authors addressed all reviewers' comments.

Reviewer 3 Report
The article may be accepted for publication.
Author Response

(The authors gave the same response as above.)
